# Factors Affecting Dental Service Utilisation in Indonesia: A Population-Based Multilevel Analysis

**DOI:** 10.3390/ijerph17155282

**Published:** 2020-07-22

**Authors:** Cornelia Melinda Adi Santoso, Taufan Bramantoro, Minh Chau Nguyen, Zsuzsa Bagoly, Attila Nagy

**Affiliations:** 1Department of Public Health and Epidemiology, Faculty of Medicine, University of Debrecen, 4028 Debrecen, Hungary; cornelia.melinda@sph.unideb.hu (C.M.A.S.); nguyen.chauminh96@gmail.com (M.C.N.); 2Department of Dental Public Health, Faculty of Dental Medicine, Universitas Airlangga, Surabaya 60286, Indonesia; taufan-b@fkg.unair.ac.id; 3Department of Laboratory Medicine, Division of Clinical Laboratory Sciences, Faculty of Medicine, University of Debrecen, 4032 Debrecen, Hungary; bagoly@med.unideb.hu

**Keywords:** dental service, social determinants, social capital, oral health behaviour, adult, Indonesia

## Abstract

This study aimed to examine the prevalence of dental service utilisation in Indonesia and its association with social determinants at individual and community levels. Cross-sectional data from the 2014 Indonesian Family Life Survey (IFLS-5) was analysed. Individual independent variables included age, sex, marital status, educational attainment, economic status, health insurance, dental pain, self-reported mouth ulcers, self-rated health status, unmet healthcare needs and smoking status, while community independent variables included cognitive, structural social capital and residential area. Multilevel logistic regressions were performed to explore the associations between independent variables at different levels and the outcome of dental service utilisation. Of the total sample of 16,860 adults aged 15 years or older in our study, around 86.4% never visited a dentist. Dental service utilisation was associated with older age, female, currently not married, higher education level and economic status, health insurance, dental pain, self-reported mouth ulcers, met healthcare needs, never smoking, living in urban areas and communities with high structural social capital. Both individual and broader social determinants influenced dental service utilisation in Indonesia. These factors should be considered in the formulation of oral health policies and programmes aiming to improve dental service utilisation in the country.

## 1. Introduction

Poor oral health is a major public health concern in Indonesia, with approximately 89% and 74% of the population having suffered from caries and periodontitis, respectively [1]. One of the most important measures to maintain good oral health is regular dental visits, as it allows the dentist to assess the risk of oral health problems, and provide preventive and restorative care [2]. A national health survey in 2018 estimated that 96% of Indonesians did not visit a dentist within the past year [1]. Several factors that influence dental service utilisation have been identified in other countries, such as age, gender, marital status, residential area, education, income, health insurance and individual health needs, including dental pain and self-perceived oral health [3,4,5].

Beyond the individual factors, the use of dental services has also been linked to broader social determinants, such as social capital, referring to attributes of social structure that facilitates the actions of individuals [2,6]. There has been increasing evidence for the influence of social capital on oral health [7,8,9,10,11]. Two dimensions of social capital include the structural (measured by social network, civic engagement) and cognitive (measured by trust, reciprocity) dimensions [12,13]. Enhanced social capital indicates a better ability to secure health-promoting resources, facilitates information sharing, including the availability of healthcare providers in the neighborhood, and reinforces the influence of peers on healthcare use [11,14,15,16]. Indonesia is a country that has a long tradition of community involvement [17], providing a unique context to study the influence of social capital on healthcare use. Delivery of public health services, including health education, is usually carried out within communities, involving community organisations.

Although there have been many studies on dental service utilisation worldwide, research conducted in Indonesia is still scarce. Moreover, most of the existing studies have only addressed individual factors. There is still limited evidence for the association between broader social determinants, such as social capital, and dental service utilisation. Identification of both individual and broader social determinants is essential to develop more effective programmes and policies that can improve dental service utilisation, reducing disparities in oral health.

The Indonesian Family Life Survey (IFLS) is a large-scale, comprehensive, and multi-purpose survey designed to study behaviours and outcomes of the Indonesian population. It is an ongoing longitudinal survey that was initiated in 1993, with follow-up data collection in 1997, 2000, 2007 and 2014. The fifth wave of the survey was the first to assess dental service utilisation among Indonesian adults. The survey collected detailed data on individual respondents, their households and the communities where they lived. Having such rich information, the survey can facilitate studies of the effects of social determinants at different levels on the population [18].

The aims of this study were to examine the prevalence of dental service utilisation in Indonesia and its association with social determinants at both individual and community levels, as well as exploring how these factors differed between urban and rural areas. We hypothesised that social determinants at both levels influenced the utilisation of dental services in Indonesia.

## 2. Materials and Methods

### 2.1. Data Source

This study was a secondary data analysis of the 2014 IFLS-5. The dataset was publicly available on the Research and Development (RAND) Corporation website [19]. The survey employed a multistage stratified sampling design. The original sampling frame included households from 13 provinces, representing 83% of the Indonesian population, while the follow-up survey was conducted on the original samples and split-off households. More detailed methodology of the survey has been described elsewhere [18].

### 2.2. Study Samples

IFLS-5 included a total of 50,148 individuals from 16,204 households. Dental service utilisation was only assessed for individuals aged 15 years or older, and the sample size was 31,416. Only individuals living in the communities that had community data on both civic engagement and social trust modules were included. After further excluding those with missing values for other covariates, the final sample in this study was 16,860 individuals within 310 communities.

### 2.3. Measurement

#### 2.3.1. Outcome Variable

The dependent variable was dental service utilisation, assessed by the question “When did you last have your teeth checked?” Respondents could answer with the following response options: month and year of their last examination, “do not know”, or “never”. Except for those who answered “never”, respondents were asked with a follow-up question “How regularly do you have your teeth checked?” The response options were “regularly” or “irregularly”. Based on the answers from these two questions, participants were classified as non-users, irregular users, and regular users.

#### 2.3.2. Individual-Level Independent Variables

Sociodemographic variables included age, sex (male or female), marital status (currently married or not married), educational attainment (elementary school or below, junior high school, senior high school, higher education), and household economic status. Principal component analysis (PCA) was used to summarise the variables concerning household assets and characteristics (ownership of house occupied by the household, other houses/buildings, land, vehicles, savings/deposits/stocks, receivables, jewellery, access to pipe water, toilet, the use of electricity/gas stove) and create the economic status scores. The first principal component was assumed to represent economic status. The resulting score was then ranked and divided into five equal groups, from quintile 1 (poorest) to quintile 5 (richest) [20,21].

Health insurance ownership was recorded as ‘yes’ or ‘no’. Dental pain and mouth ulcers were self-reported based on respondents’ experience in the past four weeks (yes or no). Self-rated health status was measured by the question “In general, how is your health?” The responses were dichotomised into unhealthy (“somewhat unhealthy”, “unhealthy”) and healthy (“somewhat healthy”, “healthy”). Unmet healthcare needs were determined based on respondents’ opinion concerning healthcare adequacy for their needs. It was classified as ‘yes’ (“less than adequate”) or ‘no’ (“just adequate”, “more than adequate”). Smoking status was assessed by asking respondents if they ever chewed tobacco or smoked cigarettes and whether they still had the habits. Based on their answers, respondents were categorised as never, former, or current smokers.

#### 2.3.3. Community-Level Independent Variables

Besides residential areas (rural or urban), two social capital variables were created in this study to represent the cognitive and structural dimensions. Items selected to create these variables were based on previous literature concerning social capital [17,22]. The community data collected by the IFLS were the responses from the community leaders [18].

Cognitive social capital was reflected by the items related to trust and safety, such as trust between different ethnicities and religions, people’s willingness to help, look out for each other, leave their children with their neighbours, ask neighbours to look after their house when they leave, general safety of the village, safety at night and necessity to be vigilant. PCA was used to summarise the responses to these indicators. The resulting score was classified into two quantiles (low, high). Similarly, structural social capital was created by summarising the existence of community programmes and activities with a PCA, which included village cooperatives, neighbourhood watch programmes, community public works, village improvements, youth and family groups, child and teen developments, elderly programmes, village savings/loans and health funds. The resulting score was then classified into tertiles (low, medium, high) [17].

### 2.4. Statistical Analysis

Descriptive analyses were performed to summarise the characteristic of the samples. Differences in the distribution of variables across the status of dental service utilisation were statistically analysed using chi-square tests. Given the nested nature of the data (individuals nested within communities), multilevel logistic regressions were performed to estimate the relationship between independent variables at different levels and outcome variable (fixed effects with random intercept). Dental service utilisation was classified into a binary variable (non-users and users).

Model 0 (null model) was first fitted without explanatory variables to report the random effects of community. Model 1 was adjusted with community variables. Model 2 was the final model, including both individual and community variables. To further explore the effect of residential area on overall results, analyses stratified by residential areas were also conducted. The significance level was set at a *p*-value < 0.05. Odds ratios (ORs) and their 95% confidence intervals (CIs) were reported. The variance partition coefficient (VPC), representing the percentage of the variance explained by the community level, was calculated using the standard logistic distribution variance of π^2^/3 (equal to 3.29), as the level 1 variance [23]. The proportional change in variance (PCV), representing the change in the community level variance between models, was also calculated. Weights were not included in our analyses. All of the analyses were performed using STATA (version 13.0, Stata Corp., College Station, TX, USA).

### 2.5. Ethical Approval

The implementation of IFLS-5 was properly reviewed and approved by the University of Gadjah Mada Research Ethics Committee in Indonesia and Institutional Review Boards of the RAND Cooperation, which was the RAND Human Subjects Protection Committee (No. s0064-06-01-CR01). Informed consent was obtained from the participants [24]. Considering that this present study was a secondary data analysis, it did not require new ethical clearance.

## 3. Results

Table 1 shows the distribution of individual and community characteristics of the respondents by the status of dental service utilisation. The proportions of non-users, irregular users, and regular users were 86.4%, 12.4%, and 1.2%, respectively. The mean (±SD) age of the respondents was 38.7 (±15.2). Most of the participants were female, married, and had educational attainment of elementary school or below. Non-users were more common among older groups, males, married individuals, current smokers, those with lower education level or economic status, not having health insurance, not suffering from dental pain or mouth ulcers, reporting unmet healthcare needs, living in rural areas, communities with high cognitive social capital, or low structural social capital. The differences in the distribution of dental service utilisation by self-rated health status were not found in our study.

Table 2 presents the multilevel logistic regression analysis of the relationship between independent variables and dental service utilisation. The percentage of variance explained by community level was 15.1% (model 0), falling to 9.5% after the inclusion of community variables (model 1), and 5.6% after the inclusion of individual and community variables. The PCV in the final model was 66.5%. Individual variables associated with dental service utilisation included age, sex, marital status, education, economic status (quintile 3–5), health insurance, dental pain, self-reported mouth ulcers, unmet healthcare needs and current smoking status, while the associated community variables included residential area and high structural social capital. The associations between self-rated health status, community cognitive social capital and dental service utilisation were not shown in our study.

Table 3 presents the multilevel logistic regression analysis of the relationship between independent variables and dental service utilisation across residential areas. Age, marital status, education, economic status, health insurance and dental pain were associated with dental service utilisation in both rural and urban areas. However, the association between economic status and dental service utilisation in urban areas was only observed for the two highest quintiles. Sex, self-reported mouth ulcers, unmet healthcare needs, current smoking status and high community structural social capital were also associated with dental service utilisation in urban areas. After the inclusion of all variables, the differences between communities could explain 6.5% of the variance in urban areas and 3.6% in rural areas.

## 4. Discussion

This study is the first to report the prevalence of dental service utilisation in Indonesia using a large sample of the general population. Moreover, it is among the few to investigate the relationships between individual and community level determinants, including social capital, and dental service utilisation using a multilevel approach. Approximately 86.4% of Indonesian adults had never been to a dentist. After adjusting for all variables, roughly 6% of the variation in the prevalence of dental service utilisation occurred at the community level and the remaining 94% was due to differences between individuals. Dental service utilisation was associated with age, marital status, education, economic status, health insurance and dental pain in both rural and urban areas, but with sex, current smoking status, self-reported mouth ulcers, unmet healthcare needs and high community structural social capital only in the urban areas.

The prevalence of non-utilisation of dental services in our study was far higher than in other countries, such as Brazil, which reported the figure to be only less than 2% [25]. Other studies investigating non-utilisation of dental services in the one year preceding the survey found the prevalence to be 33.8% in the United States [26], 60.4% in Greece [3], 63.9% in Thailand [27], and 80% in Republic of Srpska (RS), Bosnia and Herzegovina [28]. It was suggested that government investments in healthcare might influence the use of dental services in the country [29].

In line with a previous study, older adults were more likely to visit dentists than younger ones, which might be because there are more dental problems at older ages [30]. Although sex was not associated with dental service utilisation in rural areas, males in urban areas were less likely to visit a dentist than females, as confirmed by another study [31]. Females are known to pay more attention to their oral health, have more compliance and exhibit better oral health behaviours than males [32]. Our study found that married status was associated with non-utilisation of dental services, which was similar to the findings in previous studies [4,33]. An explanation might be that married people tended to be healthier than non-married people [34], having less need for health care. Another study suggested that married people might not see the benefits of dental examination unless they have needs, and thus their partners might also not encourage them to have regular examination [4].

Positive gradients between educational attainment, economic status and dental service utilisation were demonstrated, which was consistent with prior studies [25,31]. Individuals with lower education and economic status had lower oral health literacy, contributing to poor access as they might not understand the importance of oral health and their options to access dental services. [35,36]. Our finding was in agreement with previous literature that reported health insurance to be an enabling factor for dental service utilisation [25,37]. Although health insurance status in this study did not directly refer to the coverage of dental services, it could also be considered as a proxy for socio-economic status (SES). The conditions that enable individuals for having insurance may also prompt them to use dental services [25]. It was also suggested that even the availability of medical insurance increased the chance of individuals to use dental services, as presumably they could divert their savings from medical care to their dental expenses [38].

While dental pain was the only need factor associated with dental service utilisation in the rural areas, dental pain, self-reported mouth ulcers, and met healthcare needs were all associated factors in the urban areas. These findings suggested that rural dwellers had less perceived needs for dental visits than urban dwellers. Those with unmet healthcare needs in the urban areas tended not to visit the dentist. A previous study showed that dental care was the most common unmet healthcare need [39]. Our study did not show the association between self-rated health status and dental service utilisation. It was plausible that respondents’ perception of health status did not take their oral health conditions into consideration. Being a current smoker in an urban area was a factor associated with non-utilisation of dental services. A prior study found that smokers tended to visit the dentist symptomatically and not regularly for check-ups [40].

Urban dwellers were more likely to visit the dentist than rural dwellers, similarly to previous findings [30]. Less-developed public transport networks and lower availability of dental services in rural areas could be the causes [41,42]. Our study was not able to demonstrate the association between community cognitive social capital and dental service utilisation which was reported by an earlier study [43]. These different findings might be explained by the different means of measuring social capital, as the earlier study assessed social capital at an individual level.

The association between high community structural social capital and dental service utilisation was demonstrated in urban, but not rural areas. A study in China also found social capital to be more influential on health in urban areas [44]. A community with a high level of social capital, characterised by extensive social networks, organisations, and civic engagement in activities and political processes, were more likely to organise advocacy efforts to establish community health centres, partnerships with healthcare providers, and extend the reach of health services to community, including screening programmes and health education [14,22]. This mechanism could apply to oral health, as they may also be more likely to create community-based oral health promotion programmes, which eventually influence their use of dental services. The presence of organisations also provides the opportunity for social interaction, facilitating the spread of health-related information, including the availability of healthcare providers [11,15,45].

One of the limitations in our study is the cross-sectional design, which limits its ability to infer causation. Analyses were performed using unweighted data, and thus the estimates might not be generalised to the entire population in Indonesia. However, large sample size in our study increased the precision of the estimates for the associations. Self-reported data might be susceptible to inaccuracy, but social desirability bias might be reduced by assuring the participants of the confidentiality and anonymity of the information. As our analyses were limited to the variables collected in IFLS-5, there could be potential influences of unmeasured confounders, such as dental caries, periodontitis and oral submucous fibrosis. Future studies that include more indicators of oral health status, literacy, reasons of non-utilisation (i.e., fear, cost, distance, transportation, time constraints), and community variables such as dentist-to-population ratio will allow deeper exploration of the association with dental service utilisation. Longitudinal studies are needed to confirm the potential role of social capital on dental service utilisation.

## 5. Conclusions

The prevalence of dental service utilisation in Indonesia is low and it is associated with social determinants at both individual and community levels. The influences of these determinants, however, might differ between urban and rural areas. The findings of this study signalled an urgent need to improve dental service utilisation in the country. Oral health policies and programmes should be developed taking these social determinants into account and be designed to suit community-specific settings.

## Figures and Tables

**Table 1 ijerph-17-05282-t001:** Distribution of individual and community characteristics of the study population by the status of dental service utilisation.

Variables	Total Samples	Non-Users	Irregular Users	Regular Users	*p*-Value
	N (%)	N (%)	N (%)	N (%)	
-	N = 16,860	-	-	-	*-*
Dental service utilisation					
Non-users	14,573 (86.4)	-	-	-	*-*
Irregular users	2093 (12.4)	-	-	-	*-*
Regular users	194 (1.2)	-	-	-	*-*
*Individual-level variables*					
Age					
14–21	2366 (14.0)	1979 (83.6)	354 (15.0)	33 (1.4)	**<0.001**
22–34	5129 (30.4)	4460 (87.0)	601 (11.7)	68 (1.3)	
35–44	3635 (21.6)	3189 (87.7)	408 (11.2)	38 (1.1)	
45–64	4735 (28.1)	4065 (85.9)	622 (13.1)	48 (1.0)	
≥65	995 (5.9)	880 (88.4)	108 (10.9)	7 (0.7)	
Sex					
Male	8267 (49.0)	7245 (87.6)	940 (11.4)	82 (1.0)	**<0.001**
Female	8593 (51.0)	7328 (85.3)	1153 (13.4)	112 (1.3)	
Current marital status					
Married	12,287 (72.9)	10,768 (87.6)	1393 (11.4)	126 (1.0)	**<0.001**
Not married	4573 (27.1)	3805 (83.2)	700 (15.3)	68 (1.5)	
Educational attainment					
Elementary school or below	6251 (37.1)	5781 (92.5)	444 (7.1)	26 (0.4)	**<0.001**
Junior high school	3497 (20.7)	3122 (89.3)	356 (10.2)	19 (0.5)	
Senior high school	5314 (31.5)	4440 (83.5)	796 (15.0)	78 (1.5)	
Higher education	1798 (10.7)	1230 (68.4)	497 (27.6)	71 (4.0)	
Economic status					
Quintile 1 (poorest)	3401 (20.2)	3174 (93.3)	213 (6.3)	14 (0.4)	**<0.001**
Quintile 2	3423 (20.3)	3092 (90.3)	304 (8.9)	27 (0.8)	
Quintile 3	3453 (20.5)	3021 (87.5)	396 (11.5)	36 (1.0)	
Quintile 4	3383 (20.0)	2802 (82.8)	539 (15.9)	42 (1.3)	
Quintile 5 (richest)	3200 (19.0)	2484 (77.6)	641 (20.0)	75 (2.4)	
Health insurance					
Yes	8352 (49.5)	6961 (83.4)	1253 (15.0)	138 (1.6)	**<0.001**
No	8508 (50.5)	7612 (89.5)	840 (9.9)	56 (0.6)	
Dental pain					
Yes	2514 (14.9)	1847 (73.5)	619 (24.6)	48 (1.9)	**<0.001**
No	14,346 (85.1)	12,726 (88.7)	1474 (10.3)	146 (1.0)	
Self-reported mouth ulcers					
Yes	3036 (18.0)	2478 (81.6)	503 (16.6)	55 (1.8)	**<0.001**
No	13,824 (82.0)	12,095 (87.5)	1590 (11.5)	139 (1.0)	
Self-rated health status					
Healthy	13,214 (78.4)	11,421 (86.4)	1634 (12.4)	159 (1.2)	0.453
Unhealthy	3646 (21.6)	3152 (86.4)	459 (12.6)	35 (1.0)	
Unmet healthcare needs					
Yes	3593 (21.3)	3256 (90.6)	321 (8.9)	16 (0.5)	**<0.001**
No	13,267 (78.7)	11,317 (85.3)	1772 (13.4)	178 (1.3)	
Smoking status					
Never	10,380 (61.6)	8828 (85.0)	1418 (13.7)	134 (1.3)	**<0.001**
Former	859 (5.1)	726 (84.5)	119 (13.9)	14 (1.6)	
Current	5621 (33.3)	5019 (89.3)	556 (9.9)	46 (0.8)	
*Community-level variables*					
Residential area					
Urban	9255 (54.9)	7625 (82.4)	1475 (15.9)	155 (1.7)	**<0.001**
Rural	7605 (45.1)	6948 (91.4)	618 (8.1)	39 (0.5)	
Cognitive social capital					
Low	8489 (50.3)	7276 (85.7)	1106 (13.0)	107 (1.3)	**0.018**
High	8371 (49.7)	7297 (87.2)	987 (11.8)	87 (1.0)	
Structural social capital					
Low	6488 (38.5)	5839 (90.0)	586 (9.0)	63 (1.0)	**<0.001**
Medium	5972 (35.4)	5163 (86.5)	749 (12.5)	60 (1.0)	
High	4400 (26.1)	3571 (81.2)	758 (17.2)	71 (1.6)	

Note: Differences in the distribution of variables across the status of dental service utilisation were statistically analysed by chi-square tests. Bold values represent significant differences (two-sided *p*-value < 0.05).

**Table 2 ijerph-17-05282-t002:** Individual and community characteristics associated with dental service utilisation.

Variables	Model 0 ^a^	Model 1 ^b^	Model 2 ^c^
**Fixed-effects (OR with 95% CI)**	-	-	-
*Individual-level variables*	*-*	*-*	*-*
Age	-	-	**1.007 (1.003–1.011)**
Sex (ref: female)	-	-	-
Male	-	-	**0.844 (0.735–0.969)**
Current marital status (ref: not married)	-	-	-
Married	-	-	**0.718 (0.643–0.801)**
Educational attainment (ref: elementary school or below)	-	-	-
Junior high school	-	-	**1.449 (1.236–1.698)**
Senior high school	-	-	**2.065 (1.780–2.396)**
Higher education	-	-	**4.189 (3.532–4.969)**
Economic status (ref: quintile 1, poorest)	-	-	-
Quintile 2	-	-	1.207 (0.999–1.458)
Quintile 3	-	-	**1.414 (1.173–1.704)**
Quintile 4	-	-	**1.764 (1.467–2.120)**
Quintile 5 (richest)	-	-	**2.034 (1.686–2.453)**
Health insurance (ref: no)	-	-	-
Yes	-	-	**1.302 (1.177–1.440)**
Dental pain (ref: no)	-	-	-
Yes	-	-	**3.320 (2.959–3.726)**
Self-reported mouth ulcers (ref: no)	-	-	-
Yes	-	-	**1.284 (1.143–1.443)**
Self-rated health status (ref: unhealthy)	-	-	-
Healthy	-	-	0.963 (0.852–1.089)
Unmet healthcare needs (ref: no)	-	-	-
Yes	-	-	**0.819 (0.713–0.941)**
Smoking status (ref: never)	-	-	-
Former	-	-	1.048 (0.828–1.327)
Current	-	-	**0.858 (0.737–0.999)**
*Community-level variables*	*-*	*-*	*-*
Residential area (ref: rural)	-	-	-
Urban	-	**2.118 (1.748–2.565)**	**1.475 (1.249–1.742)**
Cognitive social capital (ref: low)	-	-	-
High	-	1.002 (0.845–1.188)	1.048 (0.905–1.212)
Structural social capital (ref: low)	-	-	-
Medium	-	**1.238 (1.004–1.527)**	1.173 (0.981–1.404)
High	-	**1.679 (1.339–2.104)**	**1.449 (1.194–1.759)**
**Random effects**	-	-	-
Community random variance (SE)	0.585 (0.068)	0.346 (0.046)	0.196 (0.034)
VPC	0.151	0.095	0.056
PCV (%)	Reference	40.9%	66.5%

Note: Results were based on multilevel logistic regression analyses, with non-users as a reference group. Bold values represent significant associations. The significance level was set at a *p*-value < 0.05. OR, odds ratio; CI, confidence interval; SE, standard error; VPC, variance partition coefficient; PCV, proportional change in variance. ^a^ Model 0 was fitted without explanatory variables. ^b^ Model 1 was adjusted with community variables. ^c^ Model 2 was adjusted with individual and community variables.

**Table 3 ijerph-17-05282-t003:** Individual and community characteristics associated with dental service utilisation stratified by residential areas.

Variables	Rural	Urban
	OR (95% CI)	OR (95% CI)
**Fixed-effects (OR with 95% CI)**	-	-
*Individual-level variables*	*-*	*-*
Age	**1.007 (1.001–1.014)**	**1.006 (1.001–1.011)**
Sex (ref: female)	-	-
Male	0.881 (0.679–1.144)	**0.826 (0.701–0.972)**
Current marital status (ref: not married)	-	-
Married	**0.680 (0.558–0.829)**	**0.739 (0.648–0.844)**
Educational attainment (ref: elementary school or below)	-	-
Junior high school	**1.630 (1.253–2.119)**	**1.330 (1.089–1.624)**
Senior high school	**2.389 (1.853–3.081)**	**1.863 (1.550–2.239)**
Higher education	**5.292 (3.936–7.114)**	**3.688 (2.990–4.550)**
Economic status (ref: quintile 1, poorest)	-	-
Quintile 2	**1.365 (1.037–1.797)**	1.059 (0.815–1.376)
Quintile 3	**1.788 (1.351–2.365)**	1.194 (0.926–1.540)
Quintile 4	**1.903 (1.434–2.526)**	**1.588 (1.237–2.040)**
Quintile 5 (richest)	**2.407 (1.789–3.240)**	**1.775 (1.380–2.283)**
Health insurance (ref: no)	-	-
Yes	**1.298 (1.088–1.548)**	**1.302 (1.151–1.473)**
Dental pain (ref: no)	-	-
Yes	**4.020 (3.309–4.883)**	**3.012 (2.611–3.474)**
Self-reported mouth ulcers (ref: no)	-	-
Yes	1.114 (0.899–1.381)	**1.374 (1.195–1.580)**
Self-rated health status (ref: unhealthy)	-	-
Healthy	0.882 (0.710–1.096)	1.001 (0.863–1.161)
Unmet healthcare needs (ref: no)	-	-
Yes	0.872 (0.689–1.102)	**0.784 (0.660–0.931)**
Smoking status (ref: never)	-	-
Former	0.988 (0.627–1.556)	1.073 (0.814–1.414)
Current	1.041 (0.789–1.375)	**0.774 (0.645–0.930)**
*Community-level variables*		
Cognitive social capital (ref: low)	-	-
High	1.140 (0.910–1.427)	1.019 (0.842–1.232)
Structural social capital (ref: low)	-	-
Medium	1.139 (0.889–1.458)	1.212 (0.941–1.561)
High	1.029 (0.726–1.458)	**1.600 (1.243–2.059)**
**Random effects**		
Null model community random variance (SE)	0.283 (0.071)	0.434 (0.066)
Community random variance (SE)	0.122 (0.047)	0.227 (0.044)
VPC	0.036	0.065
PCV (%)	56.9%	47.7%

Note: Results were based on multilevel logistic regression analyses, with non-users as a reference group. Bold values represent significant associations. The significance level was set at a *p*-value < 0.05. OR, odds ratio; CI, confidence interval; SE, standard error; VPC, variance partition coefficient; PCV, proportional change in variance.

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
