# Peer review of "Factors Affecting Dental Service Utilisation in Indonesia: A Population-Based Multilevel Analysis"

_ijerph, 2020, doi:10.3390/ijerph17155282_

Round 1

Reviewer 1 Report

Dear authors, the present study entitled “Factors affecting dental service utilisation in Indonesia: A population-based multilevel analysis” is an interesting article with important social finds for the dental community in Indonesia. And could help to increase de dental system in this country. However, as a reviewer, I have some concerns about the present study. Please, find below.

It was possible to find a few English language mistakes.

Introduction:
Lines 35-36: “Smokers also have different pattern of dental service utilization from non-smokers” – What is the relevance of this sentence in the first paragraph. The authors should remove or explain it.

Lines 57-59: “The main aim” and “the second aim” – Please, rewrite the sentence and join your primary and secondary objective. You should separate it only in your methods.

Material and Methods

Lines 62-73: The reviewer didn’t find the details from the Ethical Committee approval. It doesn’t matter if those data are from the internet.
The authors need to provide the features and number of the Ethical Committee.
The present study is a retrospective study using human data.

Lines 82-85: Does the questionnaire provide by the authors? Which information was possible to collect on those?

Line 93: “The resulting index was then categorised into five quintiles, from poorest to richest.”. Is this information provided by the Indonesian government? If it is a negative answer. How can you provide those statuses for the people?

Line 108 and 114: Please, provide more information about how you classified those quintiles”.

Results

Tables 2 and 3 are very confused.
Please, provide some graphics to reduce the data from those tables and for a better understanding of your study.

Discussion
Lines 242-245 : “This study is the first to report the prevalence of dental service utilisation in Indonesia using a representative sample of the population. Moreover, it is among the few to investigate the relationships between individual, community level determinants, including social capital, and dental service utilisation using multilevel approach.” - This sentence should be the first sentence of your Discussion.

Conclusion

The conclusion is not answering the objective. Please, rewrite.

Reviewer 2 Report

The manuscript titled "Factors affecting dental service utilisation in Indonesia: A population-based multilevel analysis" presents an interesting view on the dental utilization in Indonesia. There are still small things that would improve the manuscript.

On page 2, lines 83-85: Please provide more information about, how dental utilization was asked or what questions were used?

On page 3, lines 95-97: Please provide more information about how other individual variables were questionned? 

The manuscript should include more discussion about the fact that 86% of the Indonesian adults have never visited a dentist. How this compares to other studies and countries?

I also recommend English editing for the text, to check the grammar.

Reviewer 3 Report

In my opinion, this paper should be considered for publication. The manuscript is interesting and presents valuable information. Although the results obtained are logical, being highly accepted that for example, a lower socioeconomic status or lower educational level constitute the most important risk factors for delayed dental treatment, the data here researched could improve dental services utilisation in Indonesia (with probably external validity to other adyacent regions from South East Asia).

Some minor changes to improve the paper:

- The introduction is too short. If data derived from IFLS-5 Survey, publicly available from RAND website, the “master” survey should be first presented in the introduction.

- Although the objectives are pertinent, and in my opinion well written, more detailed aims of the study would be advisable. In addition, in a concise manner, the authors could include their pre-specified hypotheses.

- “Sociodemographic variables included age, sex, marital status (single or married), educational attainment (elementary school or below, junior high school, senior high school, higher education), and economic status (measured by household wealth index).”

- Why authors categorized these variables in this way? My special interest is the dichotomous variable “marital status”. Directly extracted from IFLS-5 or combined by authors? Single life and/or widowhood are also considered very relevant status for medical services utilisation, both status neglected in these study.

- No data from caries or periodontal diseases? Alcohol compsumption?

- Finally, I was impressed on the scarcity of data oral medicine related. Only recurrent aphthous stomatitis (I feel strange this variable as the only under study on this important field). Please, authors should confirm if more oral diseases were extracted or if they could be studied and added (and If possible include them). I have missed  for example oral submucous fibrosis (OSF) data. This oral potentially malignant disorder is considered as the most important adverse effect of betel nut consumption, a very extended habit in South East Asia and Indonesian rural areas (its consumption was also missed). OSF affection and oral cancer development could be relevant confounders (or main causes) among patients to receive dental services in Indonesia.

Round 2

Reviewer 1 Report

The manuscript "Factors affecting dental service utilisation in Indonesia: A population-based multilevel analysis" was modified accordingly to my suggestions.

Congrats on the excellent article.

By the way, I think that the authors are using the British English language (e.g., "utilisation" instead of "utilization").

Keep safe and good luck.

#Reviewer#